# PGPR-Soil Microbial Communities’ Interactions and Their Influence on Wheat Growth Promotion and Resistance Induction against *Mycosphaerella graminicola*

**DOI:** 10.3390/biology12111416

**Published:** 2023-11-10

**Authors:** Erika Samain, Jérôme Duclercq, Essaïd Ait Barka, Michael Eickermann, Cédric Ernenwein, Candice Mazoyon, Vivien Sarazin, Frédéric Dubois, Thierry Aussenac, Sameh Selim

**Affiliations:** 1AGHYLE UP 2018.C101, SFR Condorcet FR CNRS 3417, UniLaSalle, 19 Rue Pierre Waguet, BP 30313, CEDEX, 60026 Beauvais, France; erika.samain@hotmail.fr; 2SDP, 1 Rue Quesnay, CEDEX, 02000 Laon, France; cedric.ernenwein@rovensanext.com; 3Unité Ecologie et Dynamique des Systèmes Anthropisés (EDYSAN, UMR7058 CNRS), Université de Picardie Jules Verne (UPJV), 80039 Amiens, France; jerome.duclercq@u-picardie.fr (J.D.); candice.mazoyon@outlook.fr (C.M.); frederic.dubois@u-picardie.fr (F.D.); 4Unité de Recherche RIBP EA4707 USC INRAE 1488, SFR Condorcet FR CNRS 3417, Université de Reims Champagne Ardenne, 51100 Reims, France; ea.barka@univ-reims.fr; 5Luxembourg Institute of Science and Technology, 4422 Belvaux, Luxembourg; michael.eickermann@list.lu; 6AgroStation, 68700 Aspach-le-Bas, France; vivien.sarazin@agrostation.fr; 7UP Institut Polytechnique UniLaSalle, Université d’Artois, ULR 7519, 19 Rue Pierre Waguet, BP 30313, CEDEX, 60026 Beauvais, France; thierry.aussenac@unilasalle.fr

**Keywords:** *Mycosphaerella graminicola*, *Paenibacillus* sp. strain B2, *Arthrobacter* spp., *Microbacterium* spp., PGPR co-inoculation, induced systemic resistance, growth promotion, soil microbial communities

## Abstract

**Simple Summary:**

Plant resistance inducers, such as plant-growth-promoting rhizobacteria (PGPR), are among the most important alternatives to fungicides because they employ different modes of action, conferring protection against biotic and abiotic stresses while promoting plant growth. However, the loss of their efficacy under field conditions is a subject of debate and may be attributed to the influence of environmental factors, genetic diversity and native soil microbial communities. Additionally, the inoculation of plants with PGPR can impact the complex and dynamic balance of soil microbial communities. Recently, we published the efficiency of three PGPR in single and co-inoculation to induce wheat resistance mechanisms against Septoria leaf blotch and drought stress. Here, we investigated the interactions between PGPR and soil microbial communities.

**Abstract:**

The efficiency of plant-growth-promoting rhizobacteria (PGPR) may not be consistently maintained under field conditions due to the influence of soil microbial communities. The present study aims to investigate their impact on three PGPR-based biofertilizers in wheat. We used the PGPR *Paenibacillus* sp. strain B2 (PB2), PB2 in co-inoculation with *Arthrobacter agilis* 4042 (Mix 2), or with *Arthrobacter* sp. SSM-004 and *Microbacterium* sp. SSM-001 (Mix 3). Inoculation of PB2, Mix 2, and Mix 3 into non-sterile field soil had a positive effect on root and aboveground dry biomass, depending on the wheat cultivar. The efficiency of the PGPR was further confirmed by the protection they provided against *Mycosphaerella graminicola*, the causal agent of Septoria leaf blotch disease. PB2 exhibited protection of ≥37.8%, while Mix 2 showed ≥47.9% protection in the four cultivars tested. These results suggest that the interactions between PGPR and native soil microbial communities are crucial for promoting wheat growth and protection. Additionally, high-throughput sequencing of microbial communities conducted 7 days after PGPR inoculations revealed no negative effects of PB2, Mix 2, and Mix 3 on the soil microbial community structure. Interestingly, the presence of *Arthrobacter* spp. appeared to mitigate the potential negative effect of PB2 on bacterial community and foster root colonization by other beneficial bacterial strains.

## 1. Introduction

Thousands to millions of microbial species, including fungi and bacteria, live in the rhizosphere, interact with plant roots, and are influenced by plant species and their growth stage, which is determined by root exudates [1]. These interactions can be pathogenic, neutral, or beneficial, as PGPR (Plant Growth-Promoting Rhizobacteria). PGPRs serve as biofertilizers by enhancing nutrient accessibility to plants, as biostimulators by secreting plant hormones important for plant growth, as antagonists to pathogens, or by inducing systemic resistance (ISR) in the plant. PGPR-mediated ISR is akin to pathogen-induced systemic acquired resistance (SAR), bolstering the plant’s immune system against abiotic and biotic stresses. For instance, Septoria tritici leaf blotch (STB) disease in wheat, caused by *Mycosphaerella graminicola*, can lead to crop losses exceeding 40% [2,3]. ISR promotes the formation of physical barriers such as callose and lignin, the synthesis of protective compounds in plants, including reactive oxygen species, phytoalexins, and phenolic compounds [4,5]. Consequently, PGPR-mediated ISR is regarded as one of the most promising alternatives to fungicides, which are a subject of environmental and health-related controversies [6].

However, PGPR-induced ISR represents a quantitative, non-specific resistance with a broad-spectrum effect, which has often been demonstrated as highly effective under controlled conditions, but is not consistently maintained under field conditions [6,7]. Notably, almost all studies conducted under controlled conditions are limited in their scope, even when conducted under sterile conditions that restrict the range of strains and plant genotypes studied. Factors such as environmental conditions, genetic diversity, and the influence of native microbial communities on PGPR may provide possible explanations for the reduced efficiency of PGPR in field conditions [8]. Indeed, the soil can influence PGPR colonization, primarily through its physiochemical and existing microbial communities. These communities can have direct effects, such as trophic competitions and antagonistic or synergistic interactions, and indirect effects on plant growth and root exudation [5,9,10,11]. Inoculated PGPR must be competent, survive, proliferate, and act efficiently on plant growth and protection [12]. Native microbial communities often form competitive environments with a wide range of species, which can impact the viability and characteristics of PGPR inoculants as biofertilizer and biocontrol agents [8]. The successful colonization of this environment relies on the rhizosphere competence of the PGPR inoculum [13]. Therefore, understanding the population dynamics of the microbial community is crucial when applying microbial inoculants to establish links between plant growth promotion and protection against pathogens. In contrast to the effects of microbial communities on PGPR when used as biocontrol products, the potential environmental consequences of PGPR inoculants on native microbial communities have been relatively overlooked. Nevertheless, the introduction of high densities of viable, efficient, and competitive microbes may, at least temporarily, impact the complex and dynamic balance of soil microbial communities and the composition of taxonomic groups [14]. Ensuring the safety of microbial inoculation to the environment is a critical aspect in the development of potential biocontrol agents.

In this study, we focused on *Paenibacillus* sp. strain B2 (PB2), which is known to produce the cyclo-lipopeptide, paenimyxin, which serves as an antagonist against Gram-positive, Gram-negative bacteria, and fungi. PB2 has been shown to induce transient changes in soil bacterial community structure within 7 days of application [11,15] and to trigger ISR against *M. graminicola* in wheat [5,6]. Moreover, when used in co-inoculation with *Arthrobacter agilis* 4042 (Mix 2) or with *Arthrobacter* SSM-004 and *Microbacterium* SSM-001 (Mix 3), PB2 exhibited the ability to promote wheat growth and enhance its tolerance to drought stress, in addition to its protective effect against pathogens [10,16]. However, these previous results were obtained under controlled and sterile conditions. The objectives of our current work are to investigate the impact of soil microbial communities on the activities of PB2 when used alone or in co-inoculation regarding wheat growth and protection against *M. graminicola.* Additionally, we aim to examine the effects of these inoculations on bacterial and fungal communities in the rhizosphere and within wheat roots.

## 2. Materials and Methods

### 2.1. Microorganisms and Inoculum Preparation

*Paenibacillus* sp. strain B2 [17] was kindly provided by Dr. van Tuinen from INRA Dijon, France. *Arthrobacter agilis* was provided by C. Ernenwein of SDP, Laon, France, and *Microbacterium* sp. strain SSM1 and *Arthrobacter* sp. SSM4 were isolated in the author’s laboratory.

PGPR cultures were prepared as described by Selim et al. [11]. Briefly, to prepare the final bacterial inocula, cells were harvested at OD_0.5_, centrifuged at 2655× *g* for 10 min at 4 °C, washed twice, and then suspended in a sterile solution of 10 mM MgSO_4_ (Sigma, St. Louis, MO, USA). Bacterial cell vitality was confirmed by plating 100 µL of the inoculum on Luria-Bertani (LB, Sigma, St. Louis, MO, USA). Spores of *M. graminicola* strain IPO 323 (provided by Dr. F. Suffert, INRA Grignon) were collected from liquid cultures via centrifugation at 2655× *g* for 5 min at 15 °C, washed twice with sterile distilled water, and suspended in 10 mM MgSO_4_ (Sigma, St. Louis, MO, USA) containing 0.1% Tween 20 surfactant. Fungal spore vitality was assessed by plating 100 µL of the inoculum on potato dextrose agar (PDA, Sigma, St. Louis, MO, USA).

### 2.2. Soil Physicochemical Analysis

The soil used in this study was collected from the top 30 cm layer of an agricultural field at Institut Polytechnique UniLaSalle, Beauvais, France. This soil is a silt-loam soil (granulometric composition: silt 68.9%, clay 20.2%, sand 8.9%). Its composition includes organic matter at 1.8%, limestone at 0.2% and it has a pH of 7.1. Additionally, its nutrient content is as follows: 0.99 g N·kg^−1^, 119 mg P_2_O_5_·kg^−1^, and 194 mg K_2_O·kg^−1^.

### 2.3. Plant Material and Growth Conditions

Four wheat cultivars, Alixan, Altigo, Cellule, and Hyfi, with different degrees of resistance to STB, 4, 5.5, 6.5, and 7, respectively, on a scale of 1 (totally susceptible) to 9 (totally resistant; [18]) were used. The wheat seeds were disinfected according to Samain et al. [5], with some modifications. This involved an overnight incubation in a solution of the antibiotics oxytetracycline, streptomycin, penicillin, and ampicillin (100 mg·L^−1^ each). The seeds were then submerged in a 10% calcium hypochlorite solution for 10 min and washed three times in sterile Milli-Q water after each disinfection step. The sterilized seeds were pre-germinated on a 0.5% (*w*/*v*) water agar medium and incubated for 24 h at 4 °C, followed by 48 h at 20 °C, and finally, 24 h at 4 °C in the dark. The germinated seeds were then transferred to an inoculum consisting of a single PGPR strain or a mixture containing an equal amount of each PGPR strain, resulting in a final concentration of 10^6^ CFU·mL^−1^ in 10 mM MgSO_4_. One mL of the inoculum was added per grain for one hour with gentle shaking. For the non-inoculated condition, seeds were submerged in 10 mM MgSO_4_. After inoculation, the seeds were transferred into 250 mL pots filled with either sterile or non-sterile soil, which was a mixture of silt-loam soil and sand in a 1:1 ratio (*v*/*v*). The pots were placed in a phytotron with conditions set at 18 °C (+/−2 °C), 40% humidity, and a 16 h photoperiod with a photon flux density of 185 μmol m^−2^ s^−1^ provided by white fluorescent tubes (Philips Master Cool White 80 W/865, Lamotte Beuvron, France). The plants were irrigated three times per week with 50 mL of distilled water per pot. At the 3-leaf growth stage, the plants were inoculated with 1 mL/plant, which contained 10^6^ spores of *M. graminicola* strain IPO323 and 0.1% (*v*/*v*) tween 20.

### 2.4. Wheat Resistance Induction against M. graminicola and Wheat Biomasses

Seventeen days after infection, the leaves were collected and freeze-dried to evaluate the protective effect against *M. graminicola* in response to the inoculation with PB2 or PGPR mixtures. To quantify the infection level of *M. graminicola*, we conducted DNA extraction and qPCR analysis following the method described by Selim et al. [2]. Briefly, DNA was extracted using the DNeasy 96 Plant Kit (Qiagen, Germantown, MA, USA) according to the manufacturer’s protocol. To quantify the infection level of *M. graminicola*, we used specific primers and a TaqMan Minor Groove Binder probe targeting a 63-bp fragment of the *M. graminicola* β-tubulin-specific gene (For; GCCTTCCTACCCCACCATGT, Rev; CCTGAATCGCGCATCGTTA, Probe; FAM-TTACGCCAAGACATTC-MGB, GeneBank accession number AY547264; [19]). TaqMan^®^ assays were carried out using 12.5 μL Universal TaqMan^®^ PCR Master Mix (Life Technologies SAS, Villebon sur Yvette, France), 0.3 μM of each primer, 0.2 μM probe, 200 ng DNA, and water to a volume of 25 μL. The qPCR conditions were set as follows: 10 min at 95 °C, followed by 40 cycles of 15 s at 95 °C, and 1 min at 60 °C. All qPCR experiments were performed using the StepOnePlus Real-Time PCR System (Applied Biosystems, Waltham, MA, USA). To calibrate the qPCR analysis, we used serial dilutions of the cloned target sequence of the β-tubulin gene of *M. graminicola*, ranging from 10^2^ to 10^7^ copies, as previously described [3]. The results were expressed as the β-tubulin copy number per 100 ng of leaf DNA (BCN_100ng_). In addition, we calculated the dry biomass of roots and leaves of each wheat cultivar in each modality at the 6-week-old plant stage.

### 2.5. Taxonomic Analysis of Microbial Communities

DNA was extracted from endophytic microorganisms in both the root (100 g) and rhizosphere (300 g) of wheat plants. The rhizospheric soil was obtained by harvesting the soil that surrounds plant roots after they have been removed from their growing substrate. Samples were collected from non-inoculated and PGPR-inoculated plants (PB2 alone or in co-inoculation), and from plants that were non-infested or infested with *M. graminicola*. We used the DNeasy 96 Plant kit (Qiagen, Germantown, MA, USA) for root samples and the QIAamp DNA Mini kit (Qiagen, Hilden, Germany) for soil samples, following the respective manufacturer’s protocols. As in [20], the concentration of DNA was quantified fluorometrically using the AccuBlue High Sensitivity dsDNA Quantification Kit (Biotium, Fremont, CA, USA) and a monochromator based multimode microplate reader (Infinite M1000 PRO, Tecan System, Morgan Hill, CA, USA). For the amplification of the 16S_V3–V4_ rRNA regions, we used primers Bakt_341F (5′-CCTACGGGNGGCWGCAG-3′) and Bakt_805R (5′-GACTACHVGGGTATCTAATCC-3′; [21]). To target the fungal internal transcribed spacer (ITS) region, we used primers ITS2_KYO1 (5′-CTRYGTTCTTCATCGDT-3′) and ITS2_KYO2 (5′-TTYRCTRCGTTCTTCATC-3′; [22]). Both forward and reverse primers were designed to include overhang sequences compatible with Illumina Nextera XT index sequencing adapters. We used 5 ng of DNA per sample and verified the success of the PCR via agarose gel electrophoresis with GelGreen Nucleic Acid Gel Stain as the fluorescent intercalator, and the presence of bands of the right size on the gel with a MUPID One LED Illuminator (Advance, Suisun City, CA, USA). All amplicons were quantified using the AMPure XP Beads Kit (Beckman Coulter, Carlsbad, CA, USA) and the AccuBlue High Sensitivity dsDNA Quantitation Kit. Illumina Nextera XT Index sequencing adapters were incorporated into the amplicons through PCR. The final libraries were repurified using AMPure XP beads and quantified using the AccuBlue High Sensitivity dsDNA Quantitation Kit and the Infinite M1000 PRO microplate reader. To ensure library size and the absence of primer–dimer contamination, a 1 μL sample of a 1:50 dilution of the final library was run on a Bioanalyzer DNA 1000 chip using a Bioanalyzer 2100 (Agilent Technologies, Santa Clara, CA, USA). Purified libraries were pooled at equal molarity, denatured with freshly prepared 0.2 N NaOH, diluted to 4 pM using pre-chilled hybridization buffer, spiked with 5% of a pre-made PhiX control library (PhiX control v2, Illumina, San Diego, CA, USA), and loaded into a MiSeq v2 Reagent Kit (500 Cycles PE, Illumina, USA) for sequencing in a MiSeq sequencer (Illumina).

### 2.6. Bioinformatic Analysis

Illumina sequence reads into FastQ format were analyzed with the FROGS (Find Rapidly OTU with Galaxy Solution) pipeline [23] on the Galaxy instance (v.2.3.0) of the Genotoul bioinformatics platform (http://sigenae-workbench.toulouse.inra.fr (accessed on 13 September 2022). Sequences containing ambiguous bases (N) or those lacking specific primers were removed. We identified and trimmed primer sequences that had less than 10% of differences using Cutadapt (v.1.18; [24]). Sequence clustering was performed using the SWARM algorithm (v2.1.5, [25]). This involved an initial denoising step to create fine sequence clusters with minimal differences (d = 1), followed by a second step with an aggregation distance of three. The representative sequences for each cluster or OTU (Operational Taxonomic Units) were subjected to chimera detection using the VSEARCH algorithm for subsequent removal [26]. Taxonomic identification of each OTU was performed up to the species level using the RDPClassifier and BLAST tools. These tools were applied to the non-redundant small sub-unit database from SILVA (v123) for bacterial communities and against the ITS Warcup Training Set (v2; [27]) for the fungal communities. The high-throughput sequencing of the 16S_V3–V4_ rRNA and ITS regions resulted in 10,293,301 and 3,487,769 reads, respectively.

### 2.7. Soil Microbial Metabolic Profiles

The metabolic potential of soil communities was assessed using community-level physiological profiles (CLPP) with Biolog EcoPlate (Biolog Inc., Hayward, CA, USA; [28]). Soil samples were analyzed in triplicate using an EcoPlate consisting of 31 different carbon sources and a blank control. Each well of the EcoPlate was inoculated with 100 µL of a soil suspension (prepared by mixing 1 g of fresh soil in 10 mL of physiological water), containing approximately 10^5^ colony-forming units. The plates were incubated for 196 h at 25 °C using an OmniLog^®^ system (Biolog Inc.). To measure the rate of carbon source utilization, the reduction of tetrazolium was monitored. Tetrazolium is a redox color indicator that changes from colorless to purple, its change was detected at a wavelength of 590 nm. Data were recorded at 15 min intervals over a 192-h period and were stored as OmniLog units in Biolog data analysis software (v1.7). To calculate values for each well, blank values were subtracted from each well in the plate. To make comparisons between samples, only one absorption time point at 50 h was used, as recommended in [29]. At this time point, the metabolic potential of the soil microbial communities in each sample was calculated by determining the average color development in the wells (AWCD). This calculation involved summing the optical density data for all wells and dividing the sum by 31 (the total number of substrates). To assess the functional richness of the microbial community in the soil, we counted the total number of wells in one replicate with an absorbance greater than 25 OmniLog units.

### 2.8. Data Analysis and Statistics

Each experiment was biologically replicated twice, and each experimental condition within each experiment included a minimum of five replicates, except for high-throughput sequencing, where three technical replicates were used. To test the significance of PGPR inoculation and wheat genotype on plant growth and protection against *M. graminicola*, we performed variance analysis (ANOVA) and used Tukey’s multiple range test for separating treatment means, with a significance level set at *p* ≤ 0.05.

From the OTU × sample abundance tables, we calculated taxonomic richness and Shannon indices to assess α-diversity. To ensure comparability of taxonomic richness across samples, we rarefied the number of bacterial and fungal OTUs using the *rrarefy* function from the *vegan* R package [30]. Since the dataset did not meet the assumptions of normality, all diversity indices were compared between treatments using the R package *agricolae* [31], applying a non-parametric Kruskal–Wallis one-way analysis of variance, followed by a Conover–Iman post hoc test whenever significant. *p*-values were adjusted using the Benjamini–Hochberg FDR procedure in R with the *p.adjust* function. To examine differences in species composition between communities, we employed permutational multivariate analysis of variance (PERMANOVA) with 999 permutations. We examined variations in bacterial and fungal community structures using Nonmetric Multidimensional Scaling (NMDS) based on Jaccard distances, utilizing the *metaMDS* function from the *vegan* R package. For this analysis, we selected the 50% most abundant bacterial or fungal genera with a 30% best axis fit, using the *ordiselect* function from the vegan R package.

For the CLPP analysis, since the dataset did not meet ANOVA assumptions, we compared means between treatment groups using the R package *agricolae* and a Kruskal–Wallis test (*p* < 0.05), followed by pairwise Wilcoxon rank sum tests with Holm’s p-adjust method for multiple comparisons.

All statistical tests were performed with the R software (v4.2.2, The R Foundation, https://www.r-project.org/).

## 3. Results

### 3.1. Effects of PGPR Mixture on Wheat Biomass under Field Soil

The wheat cultivars Alixan and Cellule inoculated with PB2 showed no effects on dry biomass of roots and leaves. In contrast, Hyfi and Altigo cultivars responded differently to this inoculation. Specifically, Hyfi exhibited a notable 28.3% increase in leaf biomass and a remarkable 67% increase in root biomass when compared to non-inoculated wheat. As for Altigo, while its leaf biomass remained unaffected, there was a significant 77% increase in root biomass compared to non-inoculated plants (Figure 1). Furthermore, inoculation with Mix 2 led to increases in leaf and root biomass in all wheat cultivars, except Hyfi, where only the root biomass showed a notable increase. In contrast, the use of Mix 3 resulted in a significant increase in leaf biomass, but this effect was observed only in the Cellule and Altigo cultivars (Figure 1).

### 3.2. Resistance Induction in Wheat against M. graminicola

The level of protection induced in response to root inoculation with PGPR was determined by comparing BCN_100ng_ values in the leaves of the wheat plants, which served as controls. These control plants were non-inoculated with PGPR but infested with *M. graminicola*. We considered the protection efficiency at 40% as a minimum limit to accept the PGPR’ product as a resistance inducer. In sterile soil, BCN_100ng_ values were 10903, 379, 396, and 4444 for Alixan, Altigo, Cellule, and Hyfi, respectively. In non-sterile soil, these values were 5571, 244, 242, and 2358, respectively.

With PB2 inoculation, the induced protection against *M. graminicola* showed a significant decrease in non-sterile soil compared to sterile soil for the Alixan and Altigo cultivars. For Alixan, the protection decreased from 75.2% to 46.9%, and for Altigo, it decreased from 78.4% to 37.8%. However, no significant difference was observed for the Cellule and Hyfi cultivars, with the protection efficiency exceeding 57% (Figure 2).

Regarding Mix 2, only the cultivar Hyfi displayed a significant decrease in protection, decreasing from 77.5% in sterile soil to 47.9% in non-sterile soil. However, in sterile soil, the protection efficiency ranged from 62.4% to 83%, and in non-sterile soil, it ranged from 47.9% to 63.6%, with no significant difference among cultivars (Figure 2).

In sterile soil, mix 3 exhibited high and stable protection efficiency across all cultivars, with values exceeding 73.9%. However, under non-sterile soil conditions, Altigo and Hyfi maintained protection efficiencies of over 55% without significant decreases compared to sterile soil. In contrast, Alixan and Cellule experienced significant reduction, with protection efficiencies of 33% and 20%, respectively (Figure 2).

### 3.3. Influence of PGPR Inoculation on Microbial Communities

#### 3.3.1. α-Diversity

The impact of PGPR inoculation, whether alone or in co-inoculation, on rhizospheric and endophytic bacterial communities of two wheat cultivars (Alixan and Cellule) was assessed using three PGPR modalities, PB2, Mix 2, and Mix 3, and compared with non-inoculated plants.

Permutative multivariate analysis of variance (PermANOVA) revealed that the α-diversity of the bacterial and fungal communities under investigation were primarily influenced by the type of compartment (rhizospheric vs. endophytic, *F* = 115.92, *p* < 0.001 and *F* = 544.47, *p* < 0.001, respectively, Appendix A). Among the bacterial and fungal communities, the α-diversity of these communities was found to be significantly greater in the rhizosphere, characterized by a higher diversity and species richness compared to the roots. No significant differences in species richness were observed in either rhizospheric or endophytic bacterial communities between the two cultivars or following PGPR inoculation, except for a reduction in species richness in the rhizospheric community of Alixan, which occurred after PB2 inoculation (Table 1).

The richness of fungal species in the rhizosphere was similar in the two wheat cultivars, Alixan and Cellule. While the richness of the Cellule cultivar remained unaffected by the inoculations, the richness of the Alixan cultivar was reduced by the inoculation with Mix 2 and Mix 3. Conversely, no effect of inoculation or cultivar type on endophytic species richness was detected (Table 2).

The Shannon index of the bacterial community in the rhizosphere also showed no difference between the two cultivars, whether under non-inoculated or PGPR-inoculated conditions. This similarity extended to the endophytic communities as well, except for Cellule plants inoculated with PB2, where a significantly lower index was compared to non-inoculated wheat plants (Table 1). The Shannon index of rhizospheric fungal communities was not influenced by the cultivar type or inoculations, except for those with PB2, which exhibited a lower value than in the non-inoculated situation (for both cultivar types). In the roots, this index remained unaffected by any modality (Table 2).

The bacterial communities in the rhizosphere of non-inoculated wheat were predominantly composed of *Proteobacteria*, which collectively accounted for 43% of the bacterial species in the rhizosphere. Other dominant phyla in this bacterial community included *Patescibacteria* (16%), *Acidobacteria* (15%), *Actinobacteria* (7%), *Planctomycetes* (6%), and *Chloroflexi* (5%).

The endophytic communities were primarily characterized by an enrichment of *Proteobacteria* (46%), along with notable representations of *Actinobacteria* (29%) and *Chloroflexi* (13%, Figure 3).

PermANOVA analysis clearly showed that the structure of the bacterial community was strongly associated with the community’s origin (rhizospheric or endophytic, *F* = 544.48, *p* < 0.001). Additionally, it was influenced by the cultivar (*F* = 9.68, *p* = 0.005) and, to a lesser extent, by the type of inoculation (*F* = 3.86, *p* = 0.017, Appendix A). Combinations that included both the origin of the bacterial community and the inoculation (*F* = 3.14, *p* = 0.036), as well as combinations involving all three factors (origin, inoculation, and variety, *F* = 4.06, *p* = 0.008), also showed significant effects on community structuring.

The fungal community was primarily composed of *Ascomycota*, *Basidiomycota*, and *Zygomycota*, with the cultivar type not having a significant influence. However, permANOVA analysis (Appendix A) revealed that the proportions of these different phyla were strongly influenced by the origin of this community (rhizospheric or endophytic). Inoculation with PGPR had an impact on the fungal community by increasing the presence of *Ascomycota* at the expense of *Basidiomycota*. This effect was more pronounced in co-inoculations in the rhizosphere. In the roots, a similar response was observed, but it was the *Zygomycota* that were most affected by the inoculations (Figure 4).

#### 3.3.2. β-Diversity

The non-metric multidimensional scaling (NMDS) analysis (stress = 0.126) revealed that rhizospheric bacterial communities were distributed along two axes. The first axis was characterized by the relative abundances of *Proteobacteria* genera such as *Pseudomonas*, *Pseudorhodobacter*, *Rhizobium*, *Rhodobacter*, and *Skermanella*, while the second axis was associated with the relative abundances of certain genera of *Actinobacteria* and *Patescibacteria*, evolving in opposite directions. Inoculation with PB2 primarily influenced bacterial communities in the rhizosphere along axis 1, with more pronounced effects observed in Alixan plants (Figure 5A). Inoculation with Mix 2 also impacted bacterial communities in the rhizosphere of both cultivars, primarily along axis 2. In contrast, inoculation with Mix 3 showed no significant effect on rhizospheric communities.

The distribution of endophytic bacterial communities was characterized by two axes in the NMDS ordination (stress = 0.116, Figure 5B). Axis 1 was defined by the relative abundance of various genera of *Proteobacteria*, while axis 2 was associated with *Actinobacteria* and *Patescibacteria*. Inoculation with PB2 affected the bacterial community along axis 1 when applied to the cultivar Cellule, while the community shifted along axis 2 in the case of the cultivar Alixan. Inoculations with Mix 2 and Mix 3 also led to changes in endophytic communities, with the direction of evolution along axis 1 being opposite to that observed with PB2.

In the NMDS analysis (stress = 0.265) of the endophytic fungal community structures in the rhizosphere of both cultivars, their similarity was evident (Figure 5C). Inoculation with both cultivars with PB2 slightly influenced these communities along axis 2, with the cultivar Alixan exhibiting more notable changes. Inoculation with Mix 2 also resulted in a shift along axis 2 of the ordination, with a very similar community composition between the two cultivars, characterized by the relative abundances of *Peziza*, *Spizellomycetes*, and *Trametes*. The change in relative abundances observed in Alixan, following inoculation with Mix 3, resulted in a fungal genus assemblage that resembled a mixture of the responses observed in plants inoculated with PB2 and those inoculated with Mix 2.

Conversely, the inoculation of the cultivar Cellule with Mix 3 had a pronounced impact on the rhizospheric community along axis 1. This shift was characterized by the relative abundance of the genera such as *Alternaria*, *Rhizoctonia*, and *Reddellomyces*. Examination of the endophytic communities (stress = 0.218) revealed that, despite significant differences between the two cultivars, PGPR inoculations yielded very similar communities (Figure 5D).

#### 3.3.3. Microbial Community Function

While the number of substrates metabolized by the microbial communities in the rhizosphere remained unaffected by cultivar type or bacterial inoculations (Figure 6A), the metabolic activity, as indicated by the average well color development (AWCD), was notably higher in the rhizosphere communities of the Cellulle cultivar compared to Alixan cultivar (Figure 6B). In both genotypes, inoculation with PB2 led to enhanced metabolic capabilities in the rhizospheric communities for utilizing various EcoPlate substrates. In the Alixan cultivar, this metabolic activity was unaffected by the Mix 2 inoculation, but PB2 and Mix 3 inoculations led to increased metabolic activity. These changes were characterized by the increased utilization of carbohydrates, amino acids, and amines/amides, alongside the decreased utilization of carboxylic and acetic acids (Figure 6C). In contrast, the metabolic activity of the rhizosphere communities in the Cellule cultivar was solely increased by PB2 inoculation, which was linked to an elevated degradation of carbohydrates.

## 4. Discussion

To successfully employ PGPR strains as biofertilizer and biocontrol agents, it is essential to establish a beneficial relationship between the inoculants and the plants. It is widely acknowledged that PGPR must interact with plants to influence plant physiology [32]. This interaction depends on abiotic factors, such as physicochemical characteristics of soil, and biotic factors such as native microbial communities [9].

In our previous publications [5,10,16], we observed that PB2, Mix 2, and Mix 3, in sterile soil conditions, significantly promoted growth and induced resistance, with some variability based on the cultivar. In the current study, conducted in non-sterile field soils, PGPR strains, in general, maintained their activities in promoting wheat growth and protecting against *M. graminicola*. These effects were cultivar-dependent, and specifically, the Mix 2 treatments resulted in increased dry biomass of roots and leaves across all four wheat cultivars tested. The timing of PGPR inoculation is an important factor to consider in enabling the assimilation of the inoculating microbes into the plant rhizosphere. In our previous studies, we demonstrated that an additional inoculation with these PGPR one or two weeks after the initial one did not provide any additional benefits for wheat protection or growth promotion. However, the introduced PGPR, whether through seed incubation in a bacterial suspension for one hour or via seed coating, were detected in both ecto- and endo-roots across all wheat growth stages using specific primers [5,6,10,16]. As described in several studies, introducing PGPR as early as possible into the indigenous microbial population supports their establishment and may, in part, account for the sustained activities of PB2, Mix 2, and Mix 3 under non-sterile conditions [33,34]. Furthermore, coating the seed with the endophytic PB2, AA, SSM-001, and SSM-004 bacteria fosters the development of selected communities of beneficial bacteria within the germination structures, representing one of the earliest colonization events [35].

However, our previous results in sterile soil indicated that an increase in leaf biomass was only observed following Mix 2 inoculation in Altigo and Cellule cultivars, with no increase noted with PB2 alone [5,10]. These results suggest that interactions between PGPR and microbial communities positively impact wheat growth, as confirmed by the protection conferred by PB2 (≥37.8%) and Mix 2 (≥47.9%) against *M. graminicola* across all four cultivars tested in non-sterile field soils. Positive interactions (commensalism, mutualism, and synergism) can enable the functioning of a population [36]. Moreover, our study revealed that the genotype effect observed under sterile soil for Mix 2 on wheat growth promotion and protection against *M. graminicola* did not persist in non-sterile field soils. Nonetheless, our understanding of the factors influencing the rhizosphere competence of PGPR inoculants remains limited, and only a few studies have reported positive interactions between native microbial communities and bacterial inoculants [8,37,38].

In contrast, Mix 3 exhibited a decline in its plant growth-promoting activities and offered limited protection against *M. graminicola* compared to its performance in sterile soil [16]. However, the activity of most of PGPR strains tends to decrease in non-sterile soils when compared to sterile soils, as demonstrated by Gholami et al. (2009) [39] in maize with *P. putida* strain R-168, *P. fluorescens* strain R-93, *P. fluorescens* DSM 50090, *P. putida* DSM291, *A. lipoferum* DSM 1691, and *A. brasilense* DSM 1690. Negative interactions between introduced microbial inoculants and native microbial communities may result in the exclusion of the inoculant from the indigenous community [40,41] or trigger various allelopathic events [42,43]. Additionally, native microbial communities often represent highly competitive communities that have adapted to their environment, which can influence the biofertilization and biocontrol activities of the bacterial inoculants [8]. Moreover, the efficacy of PGPR inoculation is contingent upon the indigenous microbial communities and the specific soil used [9,44]. These factors, in turn, are influenced by root morphology, the growth stage of the plant, soil physical and chemical properties, and root exudates [36].

Ideally, microbial inputs used as biostimulants or biocontrol agents should have minimal or controlled effects on the environment. This includes factors like their dispersal, persistence, microbial function, and cycling [45]. A major concern is the impact of introduced microorganisms on the existing microbiome, which may result from direct ecological interactions, such as competition or inhibition [46]. Introduced microorganisms can indirectly affect native rhizospheric microbial communities by altering plant traits and root exudation [47], and they can even influence the innate endophytic community by modulating plant responses [48]. In this study, both the rhizospheric and endophytic bacterial and fungal communities exhibited high diversity, which probably reflects the development of the plant under non-limiting conditions [20,49]. Although PGPR inoculation did not significantly impact the diversity of rhizospheric or endophytic bacterial communities, the taxonomic composition of these communities notably changed in response to PB2 inoculation. This shift in bacterial rhizospheric communities may be attributed to the production of paenimyxin by PB2 [11], which has antagonistic effects against some Gram-positive and Gram-negative bacteria and certain fungi [11,15]. In fact, PB2 influenced the structuring of rhizospheric communities, promoting a more beneficial community with specific *Proteobacteria*, including *Pseudomonas*, *Pseudorhodobacter*, *Rhizobium*, *Rhodobacter*, and *Skermanella*; these bacteria are often recognized as PGPR themselves [50,51,52,53]. Furthermore, the maintenance of high endophytic bacterial diversity in inoculated plants suggests the preservation or enhancement of specific endophytes within the plant in response to bacterial inoculation [54,55]. In our study, the taxonomic changes observed in endophytic communities were primarily due to PB2 inoculation, which redirected the community towards a higher abundance of *Proteobacteria*. Fungal diversity was slightly reduced in the rhizosphere of both cultivars upon inoculation with PB2, resulting in the promotion of saprophytic fungi, such as *Peziza* [56], some of which may also contribute to improved plant performance [57].

The presence of *Arthrobacter agilis* 4042 in combination with PB2 (Mix 2) restricted the impact of PB2 to bacterial communities only and fostered an alternative bacterial rhizospheric structure primarily characterized by methylotrophic bacteria. Methylotrophic bacteria are well-known for their positive effects on plant growth [58] and their association with abiotic stress tolerance [59]. This effect was mirrored in the endophytic bacterial community, where although diversity remained unaffected, it maintained a distribution similar to that in the non-inoculated plants. When *Arthrobacter* sp. SSM-004 and *Microbacterium* sp. SSM-001 were combined with PB2 (Mix 3), it further constrained the effect of PB2 in shaping a more favorable bacterial community. Mix 3 also impacted rhizospheric fungi, resulting in a higher abundance of some plant pathogenic fungi, such as the genera *Alternaria* and *Rhizoctonia* [60,61]. This alteration was also observed in the endophytic community, with the induction of fungal strains of the genus *Coniothyrium*. Some strains from this genus can be used to control fungal diseases [62,63,64].

Moreover, the taxonomic changes observed after the bacterial inoculations did not affect the functionality of the soil. The high diversity observed at both rhizospheric and endophytic levels suggests that functional redundancy is able to compensate for these taxonomic changes [20]. Finally, our study primarily focused on the influence of PGPR inoculation 7 days after sowing, but microbial communities also depend on the growth stage of wheat. The influence of microbial communities on PGPR activities and vice versa varies with the sampling time, soil properties, plant development stage, and microbial strains used [65]. Further analyses conducted at various stages of wheat growth would offer a better understanding of the long-term effectiveness of PGPR. Additionally, it is essential to conduct a comprehensive investigation into the underlying mechanisms of positive and negative interactions between introduced microbial inoculants and native microbial communities, which is indeed the focus of our future research projects.

## 5. Conclusions

The present study highlighted the impact of indigenous soil microbial communities on the efficacy of PGPR inoculation using pre-germinated seeds in non-sterile agricultural cropland. Inoculation with Mix 2 demonstrated positive interactions, enhancing both PGPR biofertilization and biocontrol activities. Importantly, these positive effects were observed across different wheat genotypes, unlike the outcomes with Mix 3. The inoculation of PGPR had a significant influence on shaping bacterial and fungal communities, aiming to maximize their potential benefits for both the soil and the plant. Notably, the presence of *Arthrobacter* spp. Within Mix 2 seemed to mitigate any potential negative effects of PB2, favoring root colonization by other beneficial bacteria known for their positive contributions to plant growth and defense mechanisms against abiotic stress.

## Figures and Tables

**Figure 1 biology-12-01416-f001:**
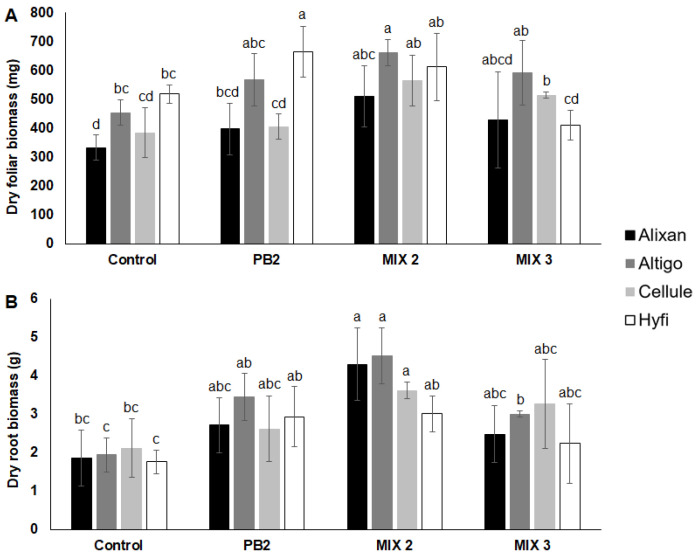
Effect of root inoculation with *Paenibacillus* sp. strain B2 (PB2), Mix 2, composed of PB2 and *Arthrobacter agilis*, or Mix 3, composed of PB2, *Microbacterium* sp. strain SSM1 and *Arthrobacter* sp. SSM4 PB2, on wheat growth in non-sterile field soils. Plants were inoculated with PGPR by dipping pre-germinated seeds in a suspension totaling 10^6^ CFU·mL^−1^. Dry biomasses of leaves (**A**) and roots (**B**) were determined 6 weeks after sowing in four wheat cultivars (Alixan, Altigo, Cellule, Hyfi). Values shown are means with SD (*n* = 5). Different lower-case letters indicate significant differences between groups according to Tukey’s test at *p* ≤ 0.05.

**Figure 2 biology-12-01416-f002:**
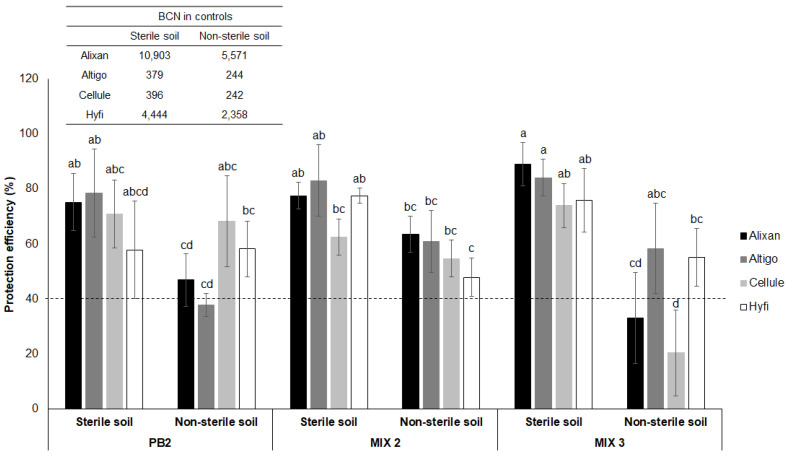
Protection efficiency induced by *Paenibacillus* sp. strain B2 (PB2), Mix 2, composed of PB2 and *Arthrobacter agilis*, and Mix 3, composed of PB2, *Microbacterium* sp. strain SSM1, and *Arthrobacter* sp. SSM4 PB2, against *M. graminicola*, strain IPO323 in non-sterile field soil, in four wheat cultivars (Alixan, Altigo, Cellule, Hyfi) at the 3-leaf growth stage, represented as percent reduction in *M. graminicola* β-tubulin copy number in 100 ng leaf DNA (BCN_100ng_ DNA) extracted from the third leaf (L3) 17 days after infection with *M. graminicola*. Plants were inoculated with Mix 2, Mix 3, and PB2 by dipping pre-germinated seeds in a suspension of 10^6^ CFU·mL^−1^. The dotted line represents the minimal protection efficiency accepted and determined by 40%. Values shown are means with SD (*n* = 5). Different lower-case letters indicate significant differences between groups according to Tukey’s test at *p* ≤ 0.05.

**Figure 3 biology-12-01416-f003:**
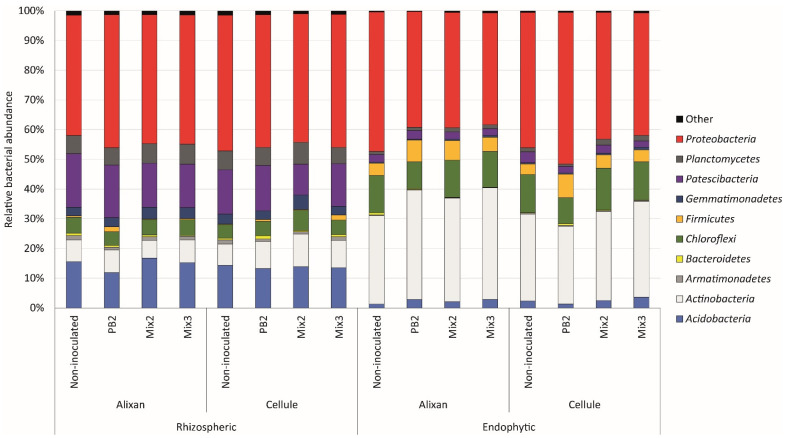
Relative abundance of bacterial 16 rRNA sequences of the different phyla for each modality. The abundance is expressed as a percentage and was calculated using the values of 16S rRNA copy numbers and nearest genome sizes for each bacterial OTU (Operational Taxonomic Unit) in the rhizosphere and endophyte compartments of Alixan and Cellule cultivars inoculated with *Paenibacillus* sp. strain B2 (PB2), Mix 2, composed of PB2 and *Arthrobacter agilis*, or Mix 3, composed of PB2, *Microbacterium* sp. strain SSM1 and *Arthrobacter* sp. SSM4 PB2, and non-inoculated control. Pre-germinated seeds were immersed in a PGPR suspension with a final concentration of 10^6^ CFU·mL^−1^. The genetic structure of the bacterial communities was observed 7 days after sowing in an agricultural field soil.

**Figure 4 biology-12-01416-f004:**
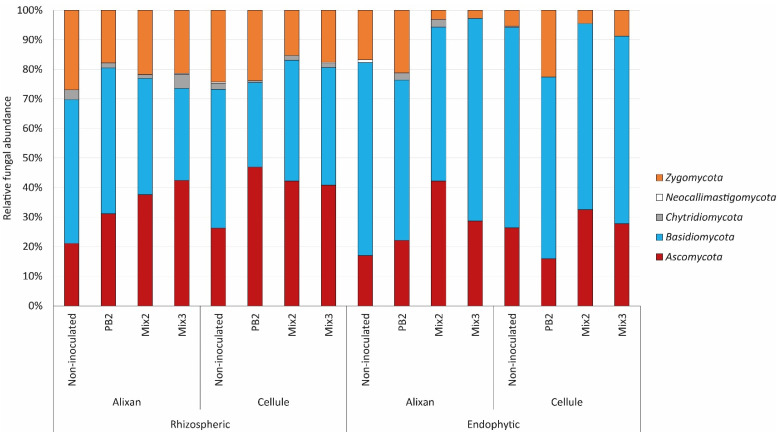
Relative abundance of fungal ITS sequences of different phyla for each modality. The abundance is expressed as a percentage and was calculated using the values of ITS copy numbers and genome sizes of the closest hits to each fungal OTU (Operational Taxonomic Unit), in the rhizosphere and endophyte compartments of Alixan and Cellule cultivars inoculated with *Paenibacillus* sp. strain B2 (PB2), Mix 2, composed of PB2 and *Arthrobacter agilis*, or Mix 3, composed of PB2, *Microbacterium* sp. strain SSM1, and *Arthrobacter* sp. SSM4 PB2, and non-inoculated control. Pre-germinated seeds were immersed in a PGPR suspension with a final concentration of 10^6^ CFU·mL^−1^. The genetic structure of the fungal communities was observed 7 days after sowing in an agricultural field soil.

**Figure 5 biology-12-01416-f005:**
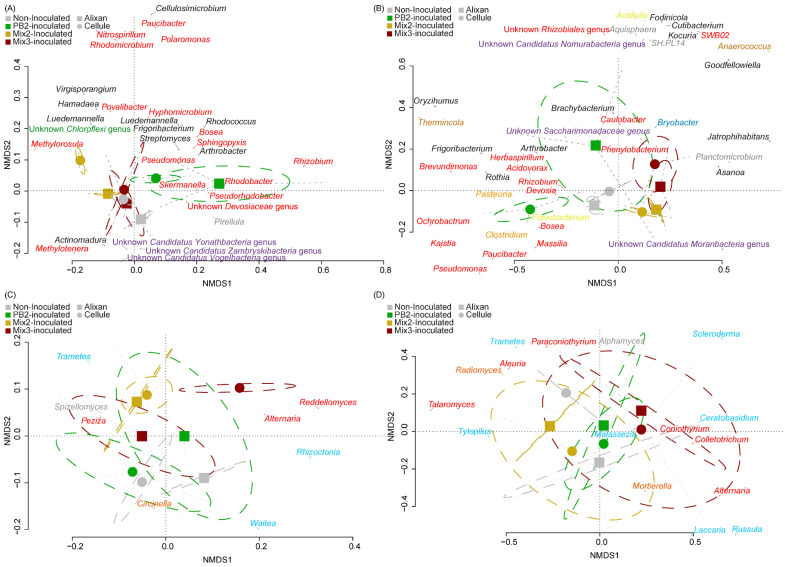
Non-metric multidimensional scaling (NMDS) of microbial communities. Rhizospheric (**A**) and endophytic (**B**) bacterial communities. The 50% most abundant bacterial genera with 30% best axis fit are shown and colored according to their phylum (*Acidobacteria* = blue, *Actinobacteria* = black, *Bacteroidetes* = yellow, *Chloroflexi* = green, *Patescibacteria* = violet, and *Proteobacteria* = red). Rhizospheric (**C**) and endophytic (**D**) fungal communities. The 50% most abundant bacterial genera with 30% best axial fit are shown and colored according to their phylum (*Ascomycota* = red, *Basidiomycota* = blue, *Chytridiomycota* = gray, and *Zygomycota* = orange). PB2, *Paenibacillus* sp. strain B2 (PB2); Mix 2, PGPR mixture composed of PB2 and *Arthrobacter agilis*; Mix 3, PGPR mixture composed of PB2, *Microbacterium* sp. strain SSM1, and *Arthrobacter* sp. SSM4.

**Figure 6 biology-12-01416-f006:**
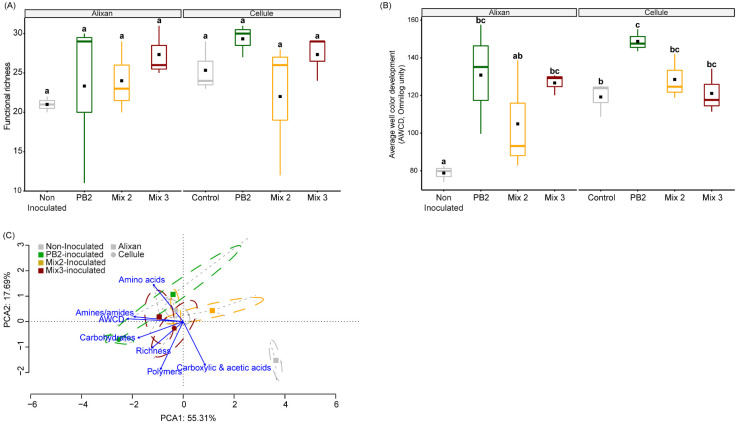
Metabolic potential of rhizospheric soil communities. Estimation of functional richness (**A**), average well color development (AWCD) expressed in OmniLog units (**B**), and principal component analysis (PCA) of carbon substrates metabolization (**C**) in the rhizospheric soil communities of Alixan and Cellule cultivars inoculated with *Paenibacillus* sp. strain B2 (PB2), Mix 2, composed of PB2 and *Arthrobacter agilis*, or Mix 3, composed of PB2, *Microbacterium* sp. strain SSM1, and *Arthrobacter* sp. SSM4 PB2, and non-inoculated control. Pre-germinated seeds were dipped in a PGPR suspension with a final concentration of 10^6^ CFU·mL^−1^. Values presented are means with SD (*n* = 3). Different lowercase letters indicate significant differences between groups according to the Kruskal–Wallis test at *p* ≤ 0.05.

**Table 1 biology-12-01416-t001:** Estimation of the richness and diversity (Shannon index) of 16S rRNA gene libraries from sequencing analysis. Pre-germinated seeds were immersed in a PGPR suspension of 10^6^ CFU·mL^−1^. The genetic structure of bacterial communities was observed 7 days after sowing in an agricultural field soil. Values shown are means with SD (*n* = 3). Different lower-case letters indicate significant differences between groups according to Kruskal–Wallis test at *p* ≤ 0.05.

Analysis Target	Cultivar	Modality	Richness			Shannon		
			Mean	SD	Sign	Mean	SD	Sign
Rhizosphere	Alixan	Non-inoculated	2454.7	222.23	b	6.435	0.01	f
		PB2	1776.7	442.10	c	6.134	0.27	bcf
		Mix2	2664.3	293.14	b	6.298	0.16	f
		Mix3	2445.0	846.29	b	6.273	0.14	cf
	Cellule	Non-inoculated	2904.3	399.74	b	6.316	0.01	f
		PB2	2735.7	225.90	b	6.330	0.11	f
		Mix2	2793.0	146.43	b	6.271	0.12	cf
		Mix3	2583.3	605.81	b	6.258	0.14	cf
Endosphere	Alixan	Non-inoculated	848.0	173.35	a	5.626	0.08	ad
		PB2	606.7	524.89	a	5.242	0.52	de
		Mix2	1069.3	178.21	a	5.922	0.08	abc
		Mix3	820.7	350.44	a	5.820	0.27	ab
	Cellule	Non-inoculated	818.5	95.46	a	5.681	0.07	a
		PB2	533.0	137.96	a	5.172	0.31	e
		Mix2	977.0	327.11	a	5.828	0.21	ab
		Mix3	670.3	422.44	a	5.649	0.35	a

**Table 2 biology-12-01416-t002:** Estimation of richness and diversity (Shannon index) of ITS gene libraries from sequencing analysis. Pre-germinated seeds were immersed in a PGPR suspension of 10^6^ CFU·mL^−1^. The genetic structure of fungal communities was observed 7 days after sowing in an agricultural field soil. Values shown are means with SD (*n* = 3). Different lower-case letters indicate significant differences between groups according to the Kruskal–Wallis test at *p* ≤ 0.05.

Analysis Target	Cultivar	Modality	Richness			Shannon		
			Mean	SD	Sign	Mean	SD	Sign
Rhizosphere	Alixan	Non-inoculated	270.67	13.32	c	4.34	0.18	d
		PB2	241.00	18.52	bc	3.22	1.29	bf
		Mix2	217.67	13.05	b	4.16	0.34	de
		Mix3	223.67	23.12	b	3.80	0.63	def
	Cellule	Non-inoculated	246.67	33.08	bc	4.37	0.06	d
		PB2	239.00	7.81	b	3.30	1.15	bef
		Mix2	233.00	8.89	b	4.16	0.15	de
		Mix3	237.67	46.69	b	4.12	0.29	def
Endosphere	Alixan	Non-inoculated	26.67	9.45	a	2.15	0.03	ac
		PB2	25.33	15.28	a	2.15	0.35	ac
		Mix2	28.33	6.35	a	2.74	0.14	ab
		Mix3	17.33	6.35	a	2.11	0.70	ac
	Cellule	Non-inoculated	23.00	8.49	a	2.55	0.42	abc
		PB2	23.67	2.31	a	1.77	0.41	c
		Mix2	28.00	8.89	a	2.61	0.29	abc
		Mix3	18.00	11.27	a	2.01	0.46	ac

## Data Availability

The raw sequence data for 16S_V3–V4_ rRNA and ITS region are available at figshare: https://doi.org/10.6084/m9.figshare.24496414 (accessed on 13 September 2022).

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
