# Peer review of "PGPR-Soil Microbial Communities’ Interactions and Their Influence on Wheat Growth Promotion and Resistance Induction against *Mycosphaerella graminicola"

_biology, 2023, doi:10.3390/biology12111416_

Round 1

Reviewer 1 Report

Comments and Suggestions for Authors

The manuscript, titled "Interactions Between PGPR and Soil Microbial Communities: Their Impact on Wheat Growth Promotion and Resistance Induction Against Mycosphaerella graminicola" delves into the evaluation of three PGPR-based biofertilizers on four distinct wheat cultivars. These PGPRs include Paenibacillus sp. strain B2 (PB2) when used alone, in combination with Arthrobacter agilis 4042 (referred to as Mix 2), or alongside Arthrobacter sp. SSM-004 and Microbacterium sp. SSM-001 (referred to as Mix 3). The study encompassed both greenhouse experiments conducted in sterilized and non-sterile field soils, and it examined the impact of microbial inoculations on wheat growth, protection against M. graminicola, and the composition of bacterial and fungal communities in the rhizosphere and within the roots of wheat plants.

Comments:

The statistical comparisons in all tables and figures need to be thoroughly reviewed and elucidated to ensure a clear understanding of the comparisons.

Have the authors considered whether a one-hour seed contact with bacterial inoculations before sowing is sufficient to observe variations in the measured parameters? Additionally, why did the authors not consider re-inoculating the soils?

In the discussion section, the authors should contemplate the significance of the timing of PGPR inoculation, as it plays a crucial role in facilitating the integration of the inoculated microbes into the plant rhizosphere.

Furthermore, there should be a comprehensive explanation of both the positive and negative interactions between the introduced microbial inoculants and the native microbial communities.

Comments on the Quality of English Language

Minor editing of English language required

Author Response

Point 1 : The statistical comparisons in all tables and figures need to be thoroughly reviewed and elucidated to ensure a clear understanding of the comparisons.

Response 1 : We employed analysis of variance (ANOVA) and Tukey's multiple range test to detect significant differences in means across various groups in a dataset, which included data on aboveground and belowground biomasses and treatment effectiveness against M. graminicola. The Tukey test is a common tool in ANOVA for comparing multiple group means and identifying statistically significant differences between specific pairs of groups. We applied the Tukey test to determine which groups showed distinct statistical differences following ANOVA, where overall significant differences were observed between the groups. Detailed information is available in the Materials and Methods section, along with F-values and p-values. Additionally, due to the dataset not meeting ANOVA assumptions, we used the Kruskal-Wallis test to evaluate the bacterial and fungal diversity indices, as well as the metabolic potential of rhizospheric soil communities. Modifications were done in Materials and Methods and in lines 363 and 389.

Point 2 : Have the authors considered whether a one-hour seed contact with bacterial inoculations before sowing is sufficient to observe variations in the measured parameters? Additionally, why did the authors not consider re-inoculating the soils?

Response 2 : We adjusted this method in our previous publication (Samain et al., 2017), where we determined that a 1-hour incubation is sufficient, and adding bacterial inoculum one week after the initial sowing did not yield any advantages compared to sowing-time inoculation.

In samain et al., 2019, 2022a and b, we tracked the presence of PGPR in both ecto- and endo-roots using specific primers and successfully detected them at all plant growth stages, including the most mature growth stage (Flag leaf).

Point 3 : In the discussion section, the authors should contemplate the significance of the timing of PGPR inoculation, as it plays a crucial role in facilitating the integration of the inoculated microbes into the plant rhizosphere.

Response 3 : The requested specific paragraph was added in discussion part lines 655-660 [In our previous studies, we demonstrated that an additional inoculation with these PGPR one or two weeks after the initial one did not provide any additional benefits for wheat protection or growth promotion. However, the introduced PGPR, whether through seed incubation in a bacterial suspension for one hour or via seed coating, were detected in both ecto- and endo-roots across all wheat growth stages using specific primers [5, 6, 10, 16].

Point 4 : Furthermore, there should be a comprehensive explanation of both the positive and negative interactions between the introduced microbial inoculants and the native microbial communities.

Response 4 : Many thanks for this important comment, which is the focus of our upcoming project. We briefly mentioned this in the discussion's closing sentence (lines 745-749). In this article, we highlighted both the positive and negative impacts earlier in the discussion. Starting from line 673, we provided an explanation for why MIX2 outperformed PB2 alone or MIX3 based on the results of soil microbial communities. We discussed that the negative impact of PB2 could be attributed to antibiotic production and demonstrated that the presence of Arthrobacter agilis 4042 in combination with PB2 (Mix 2) limited PB2's impact to bacterial communities, leading to an alternative bacterial rhizospheric structure primarily characterized by methylotrophic bacteria. We also explained that "Methylotrophic bacteria are well-known for their positive effects on plant growth [59] and their association with abiotic stress tolerance [60]. Furthermore, we clarified why MIX 3 is less effective than MIX2, stating that when Arthrobacter sp. SSM-004 and Microbacterium sp. SSM-001 were combined with PB2 (Mix 3), it further limited PB2's effect in shaping a more favorable bacterial community. Mix 3 also impacted rhizospheric fungi, resulting in a higher abundance of certain plant pathogenic fungi, such as the genera Alternaria and Rhizoctonia [61,62]. We observed a similar alteration in the endophytic community, with the induction of fungal strains from the genus Coniothyrium, some of which can be used to control fungal diseases [63–65].

Reviewer 2 Report

Comments and Suggestions for Authors

The manuscript «PGPR-soil microbial communities’ interactions and their influence on wheat growth promotion and resistance induction against Mycosphaerella graminicola» authors Erika Samain, Jérôme Duclercq, Essaïd Ait Barka, Michael Eickermann, Cédric Ernenwein, Candice Mazoyon, Vivien Sarazin, Frédéric Dubois, Thierry Aussenac, Sameh Selim is devoted to the current problem of the effectiveness of using PGPR to stimulate plant growth and protect them from phytopathogens in artificial sterile and natural non-sterile conditions of soil microbiomes. The study was carried out at a high technological level using modern methods of molecular and metabolomic analysis. The results of the study expand the understanding of the interaction of plants and microorganisms depending on environmental factors, and also allow us to evaluate the practical value of using biological products based on PGPR.

The quality of the manuscript text is very high. The authors describe the methods used in great detail. Several questions arose while studying the manuscript:

1. Paragraph 2.6 states: “DNA was extracted from ... rhizosphere (300g) of wheat plants.” But the rhizosphere is the narrow region of soil or substrate that is directly influenced by root secretions and associated soil microorganisms. How did the authors determine the part of the soil that belongs specifically to the rhizosphere? Or would it be more accurate to say that the soil used was from the container in which the wheat plants were growing?

2. Paragraph 2.8 (lines 237-238) states: “For each experiment, we conducted a minimum of five technical replicates and two biological replicates...”. Does this mean that in each experiment no more than 10 plants were studied in the experimental variant, and in each technical repetition there were only two plants, including for assessing morphometric parameters? If this is so, then this number of plants is not enough to provide a representative assessment of trait variation. If not, then you need to indicate how many plants were evaluated in each experimental variant.

3. Paragraph 3.2 and Figure 2 show the results of studying the induction of wheat resistance to M. graminicola. Three interacting factors were studied: cultivar genotype (4 cultivars), PGPR treatment (PB2, MIX2, MIX3), and soil type (Sterile soil, Non-sterile soil). The text of paragraph 3.2 and the caption to the figure do not indicate which ANOVA analysis method was used. Was the influence of a group of factors assessed?

4. In the References section, clause 18 requires clarification of the format. There is no indication of this source in the text of the manuscript.

I believe that the manuscript can be accepted for publication in the journal Biology after minor corrections by the authors.

Author Response

Point 1. Paragraph 2.6 states: “DNA was extracted from ... rhizosphere (300g) of wheat plants.” But the rhizosphere is the narrow region of soil or substrate that is directly influenced by root secretions and associated soil microorganisms. How did the authors determine the part of the soil that belongs specifically to the rhizosphere? Or would it be more accurate to say that the soil used was from the container in which the wheat plants were growing?

Response 1: The soil described as rhizospheric corresponds to the soil that surrounds the plant roots and adheres to the roots after they have been removed from their substrate. This clarification has been added to the Materials and Methods section.

Point 2. Paragraph 2.8 (lines 237-238) states: “For each experiment, we conducted a minimum of five technical replicates and two biological replicates...”. Does this mean that in each experiment no more than 10 plants were studied in the experimental variant, and in each technical repetition there were only two plants, including for assessing morphometric parameters? If this is so, then this number of plants is not enough to provide a representative assessment of trait variation. If not, then you need to indicate how many plants were evaluated in each experimental variant.

Response 2: modifications were done to clarify this point (lines 252-254) : All experiments were repeated twice and each experiment contained at least five replicates, except for high-throughput sequencing, where three replicates were used.

Point 3. Paragraph 3.2 and Figure 2 show the results of studying the induction of wheat resistance to M. graminicola. Three interacting factors were studied: cultivar genotype (4 cultivars), PGPR treatment (PB2, MIX2, MIX3), and soil type (Sterile soil, Non-sterile soil). The text of paragraph 3.2 and the caption to the figure do not indicate which ANOVA analysis method was used. Was the influence of a group of factors assessed?

Response 3: We used simple one way ANOVA to study the difference between the two conditions of sterile and non-sterile soils for each PGPR product separately. Modifications were realized at lines 305-307 to clarify this point.

Point 4. In the References section, clause 18 requires clarification of the format. There is no indication of this source in the text of the manuscript.

Response 4: Reference [18] was added at line 135.